# On the Directivity of Acoustic Waves Generated by the Angle Beam Wedge Actuator in Thin-Walled Structures

**Sergey Shevtsov** [1,*] **, Valery Chebanenko** [1] **, Maria Shevtsova** [2] **, Evgenia Kirillova** [2] **and Evgeny Rozhkov** [3]

1   Aircraft Systems and Technologies Lab at FRC The Southern Scientific Centre of Russian Academy of Science, Rostov on Don 344006, Russia
2   Faculty of Architecture and Civil Engineering, RheinMain University of Applied Science, 65197 Wiesbaden, Germany
3   Institute of Mechanics and Applied Mathematics at the Southern Federal University, Rostov on Don 344091, Russia
*   Correspondence: sergnshevtsov@gmail.com; Tel.: +7-903-401-3385

**Abstract:** The paper aims to develop improved acoustic-based structural health monitoring (SHM) and nondestructive evaluation (NDE) techniques, which provide the waves directivity emitted by the angle beam wedge actuators in thin-walled structures made of plastic materials and polymeric composites. Our investigation includes the dispersive analysis of the waves that can be excited in the studied plastic panel. Its results allowed to find two kinds of generated acoustic waves—anti-symmetric Lamb waves (A0) and shear horizontally polarized SH waves (SS0). The bounds of the chosen frequency range for the experimental and numerical studies were accepted as a compromise between the desire to obtain a high defect resolution by generating short waves, their adjustable directivity, and maximum propagation length. The finite element model for the transducer was built by using the results of an actuator structure experimental study. The frequency response functions for the actuator current and oscillation amplitude of the footprint surface demonstrated good agreement. The found eigenfrequencies of the actuator's structure were used for the numerical and experimental study of the Lamb and SH wave generation and propagation in a thin-walled plastic panel. Our results convincingly demonstrated the satisfactory directivity of the actuated waves at their excitation on the frequencies that corresponded to the natural modes of the actuator oscillation. The authors assume that an efficient use of the proposed technique for other analyzed quasi-isotropic materials and applied actuators can be provided by preliminary research using a similar approach and methods presented in this article.

**Keywords:** acoustic based SHM; plastics and polymeric composites; Lamb waves; horizontally polarized SH waves; angle beam wedge transducer; waves directivity

## 1. Introduction

The goal of the paper is to develop an improved technique for the nondestructive evaluation (NDE) and structural health monitoring (SHM) of load-carrying thin-walled structures using direction-controlled surface acoustic waves, which are generated by the surface-mounted piezoelectric transducers. The Lamb waves are used in active SHM systems because they can propagate over large distances in the thin-walled parts, interacting directly with the potential defects that allow to estimate the state of health and reliability of aircrafts, rotorcrafts, and another machines. The most interesting benefits of using guided Lamb waves is their ability to detect defects in thin-walled structures such as

inclusions, porosity, undesirable local changes of the material's mechanical properties, and sometimes a possibility to control the waves directivity, i.e., beamsteering. The aspects of using acoustic waves in the active SHM system are outlined and considered in monographs [1–3], in papers [4–8], and in theses [9–12].

Typically, every full-featured Lamb wave-based SHM involves four levels, referred to as [5]:

- detection of the occurrence of an unsafe irregularity;
- identification of the geometric location of the irregularity;
- determination of the magnitude or severity of the irregularity;
- prognostic estimation of the remaining service life/strength.

The first phase of every acoustic SHM technique assumes the proper choice of the wave excitation frequency, its intensity, propagation distance, and directivity, which depend on the geometry of the part under investigation, on the material damping, and its structural anisotropy. Among these degrees of freedom, which should provide the selected wave motion excitation method, the directivity of wave propagation is most difficult to implement as desired. In order to excite the acoustic waves in a structure, a variety of different techniques is used, including piezoelectric transducers in the form of circular [4,6], annular tablets [13], or rectangular piezoelectric (PZT) patches [12,13], by PZT wedge transducers [11,14–16], piezoelectric wafer active (PWAS) and macro-fiber composite (MFC) actuators [17–19], and by laser beam excitation [20]. For the sensoring of the waves, which are reflected or scattered on the defects, small piezoelectric transducers or non-contact laser Doppler vibrometry (LDV) are most often used.

The distance of the acoustic wave propagation depends on the contact stress amplitude generated by the surface mounted actuator. In the most common case, the wave amplitude $A(r)$ at a distance r from the source of excitation is determined as

$$A(r) \cong \widetilde{A}/\sqrt{r}, \tag{1}$$

where $\widetilde{A}$ is the wave amplitude at the excitation source [21,22]. Along with the so-called geometric attenuation (1), in plastics and polymeric composites the wave attenuation due to material damping is observed [5,21]. This phenomenon allows to determine the studied material damping properties experimentally for the cases where an omnidirectional transducer is used, which excites the waves in the part of simple geometry and homogeneously distributed the material's properties. However, the correct localization of a possible irregularity requires an ability to control the direction of excited waves propagation that is provided only by the PWAS, MFC, and angle beam wedge actuator. The latter has the advantage that it can be moved along the surface of the part, while the PWAS and MFC should be fixed on the part's surface. In NDE and SHM practice, the angle beam wedge actuators, which use the principles of refraction and mode conversion to produce refracted shear or longitudinal waves in the test material, are very efficient for the inspection of thick metallic components as they provide a large scanning index. These abilities have been studied and confirmed by earlier [23–26] and later theoretical and experimental research [16,27,28]. The first work dedicated to the use of the wedge actuator for the wave generation in thin-walled plates was [29], where the wedge method of generating guided waves was analyzed with particular attention being focused on the relationship between the angularly dependent excitation amplitude of a given mode and the physical parameters of the transducer and wedge used to excite the mode. This theoretical investigation used the analytical approach assuming that the transducer produces a roughly parabolic pressure distribution on the contact footprint of the form

$$p(\alpha) = \begin{cases} \sigma_0\left(1 - \dfrac{\alpha^2}{(D/2)^2}\right) & if \quad |\alpha| \leq D/2 \\ 0 \quad if \quad |\alpha| > D/2 \end{cases}, \tag{2}$$

where $\sigma_0$ represents the maximum pressure which occurs at the center of the transducer face, $\alpha = 0$, and the transducer has a width $D$. More detailed later studies [16,26–28] based on the finite-element analysis established a very complex contact stress distribution that produces both out-of-plane and in-plane displacements within the elliptic footprint.

Later papers [14,15] discuss difficulties of the wave tuning at its excitation in thin-walled structures. First, the phase velocity $c_w$ (the velocity of the longitudinal or transverse acoustic waves in the wedge) must be smaller than $c_{phase}$ (the phase velocity of a desired wave mode at a selected frequency in the thin-wall structure). This can limit either the choices of wedge materials appropriate for the generation of a desired mode or the availability of modes that can be generated with a given wedge material. Secondly, spurious signals, resulting from wave reverberation inside the wedge, may deteriorate the quality of the useful signal, that is, the intended applied wave. Thirdly, due to beam spreading of acoustic waves propagating through the wedge to the structure's surface, other wave modes beyond the mode of interest may be generated. This statement is confirmed by the results of [26], where it is reported that at a large enough wedge-specimen contact area, the undesirable aperture effects can appear.

The papers [27,28] propose an analytical approach to the problem of modeling linear Rayleigh wave sound fields generated by angle beam wedge transducers. In these papers, the reciprocity theorem for the dynamics of elastic bodies is transformed into integral representations, and the fundamental solutions of wave motion equations are obtained using Green's function method. The authors show that the results obtained by the proposed technique, which neglects the waves attenuation both in the excited aluminum specimen and wedge, are more numerically stable than those obtained by the 3-D Rayleigh wave model. In order to improve the developed technique, the authors propose to take into account the leaky energy of Rayleigh waves back into the wedge from underneath it. Moreover, if such transducers are used to excite the acoustic waves in plastics and polymeric composite materials, which have a very intensive structural damping, the upper bound of the excitation frequency range is limited by the fast attenuation of wave energy and amplitude.

In this article, we present results of the numerical and experimental study of acoustic wave excitation in a square quasi-isotropic plastic plane panel with the dimensions $50 \times 50 \times 0.4$ cm surrounded by the absorbing layer. This layer is made of porous rubbery belt, which simulates a perfectly matched layer intended to minimize reflection of the traveling waves and formation of standing waves. The mechanical properties of the studied panel's material were preliminarily measured experimentally using the modified technique described in [30]. On the base of calculated elastic properties, the dispersion analysis for Lamb waves and for shear horizontally polarized waves was performed. This allowed to determine the types of the waves that can be generated in the chosen frequency range without intensive attenuation. In order to determine the properties of the used angle beam wedge transducer Olympus V414-SB-ABWS-3-45 for the subsequent use of these properties in the numerical modeling, it has been subjected to self-testing to determine the electrical capacitance, the frequency response functions (FRF) of the consumed electric current, and the amplitude of the normal displacements on the contact surface. In the frequency range of 10–100 kHz, three resonance frequencies were chosen at which the experimental and numerical studies of the generated wave field on the surface of the plastic panel were fulfilled. For two generated waves types—anti-symmetric Lamb waves and shear horizontally polarized waves (SS0)—the wave directivity was studied using the finite-element model of the system, where the wedge's dimensions and its electric and mechanical properties were fully consistent with those for the real transducer. The numerical results, obtained for the Lamb waves, were supplemented by the experimental measurements. In these experiments, small-sized piezoelectric sensors that are sensitive to the out-of-plane displacements of the excited panel surface were used. Our results show that satisfactory directivity (beamsteering) of the wavelengths up to 1–2 cm, which can be used in NDE and SHM for testing heavily damping plastic or composite materials, can be achieved by using angle beam wedge actuators at the frequencies matching the transducer's natural vibration modes, which should be preliminarily determined. These results confirm the relevance of a similar

study for the more complex orthotropic composite materials, whose anisotropy can sufficiently distort the directional characteristics of the waves generated by the angle beam wedge actuator.

## 2. The Experimental Investigation of Acoustic Waves Excited in the Plastic Panel under Study

Our experimental study has been divided into three subsections. In order to calculate the dispersion relations for the waves that can be excited in the studied panel, which is made of Polypropilen GC40S-402 (BASF Procom®), its mechanical properties were measured in a series of tests using a testing machine and further numerically processed. Used in subsequent experiments, the angle beam wedge actuator Olympus V414-SB-ABWS-3-45 was tested alone to reconstruct its characteristics, which were used in the development of its finite-element prototype. At the final stage, the angular and radial intensity distributions of the wave fields generated by this actuator in a plastic plate were investigated.

### 2.1. Determination of Elastic Properties for the Material of the Studied Panel

The specimens with the dimensions $250 \times 25 \times 4$ mm were tested by the testing machine TIRA test 2850 with an extensometer, which measured the small displacement during the stroke of the crosshead. The extensometer having a base of 100 mm was calibrated to determine a dependency between displacement and output voltage. The calibration curves obtained at the tensile loading and unloading allowed to correctly calculate a tensile strain during specimen elongation and contraction. All tests were performed within the elastic region by two to three cycles of loading–unloading with a crosshead speed of 1 mm/min. The strain and tensile force time histories were stored in text files, then those were recalculated to remove experimental noise and calculate the elastic modulus. The Poisson ratio was measured using pairs of strain gauges placed on the sample's surfaces normal to their length and calculated according to a similar technique. All obtained values of the elastic modulus were verified in the experiment of resonance frequency determination. The samples with known dimensions were fixed as cantilever and harmonically excited by the shaker at changing frequencies. The eigenfrequencies of the 1st bending vibration modes were used to calculate the Young's modulus using the classical formula for the Euler–Bernoulli beam. The observable discrepancy between the moduli, which were calculated according to the different methods, did not exceed 3.5%. A comparison of the moduli measured for the specimens carved along two perpendicular directions of the sheet of material under study proved that the material can be considered quasi-isotropic. All data for the studied specimens are summarized in Table 1.

**Table 1.** Mechanical properties of the studied plate's material.

| Young's Modulus, GPa | Poisson Ratio | Density, kg/m$^3$ |
|:---:|:---:|:---:|
| $7.4 \pm 1.1$ | $0.33 \pm 0.04$ | $1230 \pm 42$ |

### 2.2. Reconstruction of the Wedge Actuator Structure and Electro-Mechanical Properties

The present research combined two parts—an experimental and a numerical one—and the latter was fulfilled by using the finite-element approach. Therefore, it was important to determine the best possible matching between the characteristics of the simulated and the real actuator (see Figure 1), both in terms of the dimensions and the electromechanical properties.

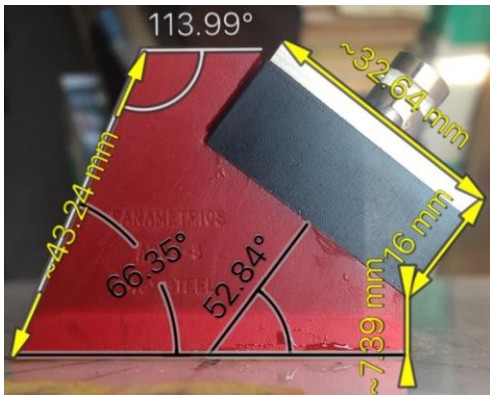

**Figure 1.** Photograph of the Olympus angle beam wedge transducer used and its dimensions.

The topology of the actuator's structure was reconstructed by using the technical notes of the developer. The properties of the active PZT element were determined using the transducer sketch, dimensions of the waveguide made of Lucite, and the measured electric capacity of the transducer. The properties of the backing, which is usually highly attenuative high-density material that is used to absorb the energy radiating from the back face of the active element, were derived from experimental measurements of the frequency response function for the consumed electric current and amplitude of normal displacements on the contact surface of the transducer. The value of the acoustic impedance of the backing was chosen to match the acoustic impedance of the active PZT element. The result is a heavily damped transducer that displays a good frequency range resolution but not a very high signal amplitude. The properties of the matching layer were chosen to serve it as an acoustic transformer between the high acoustic impedance of the active element and the waveguide wedge, which is of lower acoustic impedance. In order to simplify the finite element (FE) model of the device, the upper part of the corps was transformed to a cylindrical shape (see Figure 2).

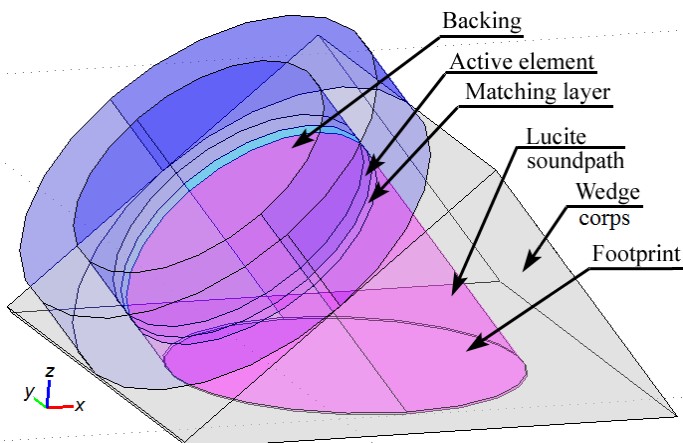

**Figure 2.** Geometry of the finite-element prototype for the used wedge transducer.

The mechanical properties for the actuator parts' materials, which were determined after performing the experiments listed above, are presented in Table 2. The values of stiffness and matrices for the active element made of PZT-5H polarized ceramics are taken from the producer's data. These matrices are used to formulate and solve the piezoelectricity equations of the FE calculations of the stress charge. In the built FE model, these matrices were defined in the tilted coordinate system that conforms to the wedge actuator's geometry.

**Table 2.** Mechanical properties of the actuator parts' materials.

| Actuator's Part | Young's Modulus, GPa | Poisson Ratio | Mass Density, kg/m$^3$ | Loss Factor |
|---|---|---|---|---|
| Body | 6.0 | 0.33 | 1500 | 0.075 |
| Backing layer | 50 | 0.1 | 7500 | 0.15 |
| Matching layer | 6.0 | 0.3 | 1500 | 0.025 |
| Lucite sound path | 4.0 | 0.33 | 1200 | 0.025 |
| Active element (PZT-5H) | The elasticity, coupling, and relative permittivity matrices contain all data | | 7500 | 0.15 |

In our numerical study of the FE prototype actuator we simulated the frequency response functions for the actuator's consumed electric current and amplitude of normal displacements on the contact surface of the transducer's model like during the experiments with the used Olympus angle beam wedge transducer. Our FE model's simulation demonstrated the global maximum of the current's FRF at the frequency ~550 kHz and three local maxima in the low frequency region: at 15, 30, and 65 kHz. Our subsequent experimental study showed that at frequencies above 100 kHz, the attenuation of acoustic waves in a plastic material of the considered type is so large that it is meaningless to study the directivity of waves at such high frequencies. Therefore, the thorough analysis of acoustic waves excited at these three frequencies has been fulfilled.

*2.3. Experimental Study on the Directivity of Acoustic Waves Excited by the Wedge Actuator in a Plastic Panel*

Due to quasi-isotropic mechanical properties of the plastic material only one quarter of the panel was studied (see Figure 3). The measurements of out-of-plane displacement amplitudes were carried out by small piezoelectric sensors. They were installed at the points that are positioned at the intersections of radial straight lines drawn from the center of the plate, with circles of radii of 10 and 17.5 cm. The angular step between adjacent radial lines was 15°. In order to cover the range of angles (0°–180°) the measurements were made in two stages: at the actuator oriented along the *x*-axis, and with the opposite orientation (see Figure 3a). The actuator's and sensors' signals were registered by the oscilloscope (LeCroy) and stored for each orientation angle in the text files for further numerical processing. To avoid the reflection of the traveling waves from the panel edges and the formation of standing waves, the studied circular area was surrounded by an absorbing layer made of porous rubber. This means was also used in the FE modeling.

In order to supply the tight contact between the panel surface, sensors, and actuator, the very viscous couplant SWC-2 (Olympus) was used.

Two different driving signals (see Figure 4) were prepared using Windows-based software ArbExpress AXW100 Waveform Creation and Editing Tool for Tektronix AWG/AFG and then formed by the wide frequency range generator Tektronix whose output is amplified by the piezodrivers PA94 (Apex Co., USA) and drives the actuator.

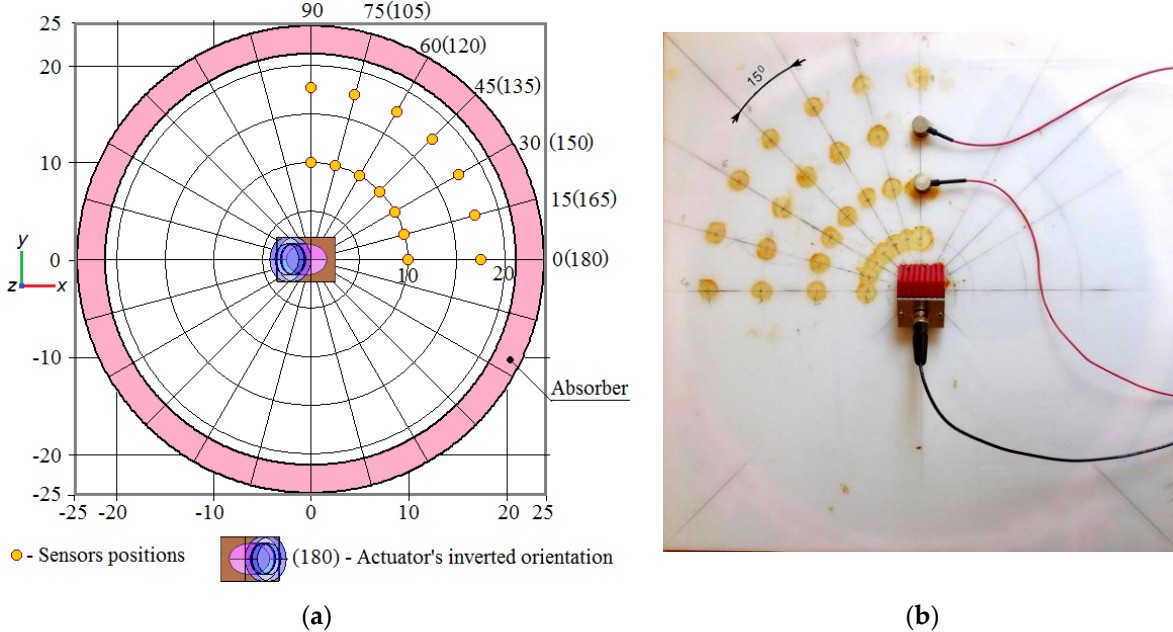

**Figure 3.** The plastic panel under study: (**a**) schematic view; (**b**) photograph of studied plastic panel with installed angle beam wedge actuator and one pair of sensors.

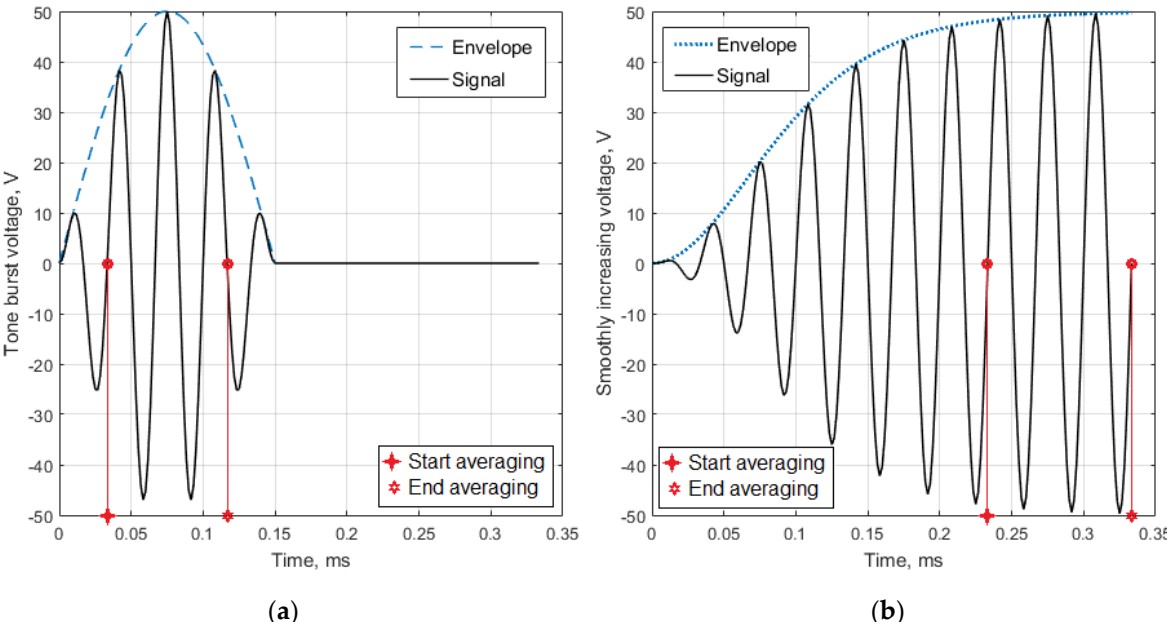

**Figure 4.** Two kinds of the actuator's driving signals used for determination of the generated wave speeds and of the actuator's directivity: (**a**) tone burst signal; (**b**) signal with a smooth increase and stabilization of the amplitude. Both signals have the carrying frequency 30 kHz.

To avoid a sharp jump in the driving potential and unwanted response of the mechanical system, these signals have zero-value first derivatives:

$$U_{TB} = Ampl \cdot \sin 2\pi f_{car} t \cdot \begin{cases} \sin \frac{2\pi f_{car}}{N_W} & at \quad 0 < t < f_{car}/2N_W \\ 0 \quad at \quad t > f_{car}/2N_W \end{cases}, \tag{3a}$$

$$U_{SI} = Ampl \cdot \sin 2\pi f_{car} t \cdot \tanh^2 t/\tau. \tag{3b}$$

Both signals (4a) and (4b) depend on the voltage amplitude *Ampl*, the carrier frequency $f_{car}$, and time *t*. The envelope of tone burst signal (4a) depends of the waves number $N_W$ in the signal, whereas the signal with the smoothly increasing and stabilizing amplitude (4b) depends on the time constant $\tau$, which determines the settling time for signal amplitude.

The wave speed was calculated on the base of time lag between two signals registered by the sensors placed on the wave path. For the assessment of directivity of excited waves, the time history of signal registered by the sensor placed at a distance of 10 cm from the center of the elliptic footprint of the actuator was processed by the averaging algorithm. This algorithm performed the integration of the signal's absolute values within the time interval, corresponding to two time instants of driving electric potential depicted in Figure 4. The right boundary of the integration interval was always chosen before the wave reached the edge of the plate.

The main experimental results are presented below together with results of the theoretical study to demonstrate their accordance.

## 3. Numerical Study of Acoustic Wave Propagation

Both experimental and theoretical studies of acoustic waves propagated in the investigated plastic panel have been fulfilled at the excitation frequencies below 100 kHz. It was due to the high attenuation of the excited waves and due to limited abilities of affordable experimental equipment. In order to understand what kinds of acoustic waves can be excited in the panel at these frequencies, a dispersion analysis of the Lamb and the horizontally polarized waves was implemented.

### 3.1. Dispersion Analysis of Acoustic Waves That Can be Excited in the Plastic Panel Under Study

Numerically obtained solution of the dispersion equations [1] for the symmetric (4a) and anti-symmetric (4b) waves are presented in Figure 5.

$$\frac{\tan(qh)}{\tan(ph)} = -\frac{4k^2 pq}{\left(q^2 - k^2\right)^2},\tag{4a}$$

$$\frac{\tan(qh)}{\tan(ph)} = -\frac{\left(q^2 - k^2\right)^2}{4k^2 pq},\tag{4b}$$

where

$$p^2 = \frac{\omega^2}{c_L^2} - k^2;\tag{5a}$$

$$q^2 = \frac{\omega^2}{c_T^2} - k^2\tag{5b}$$

$k = \omega/c$ is the wavenumber, $c_L^2 = (\lambda + \mu)/\rho$ and $c_T^2 = \mu/\rho$ are the pressure (longitudinal) and shear (transverse) wavespeeds, $\lambda$ and $\mu$ are the Lame constants, *h* is the half-thickness of the elastic layer, and $\rho$ is the mass density. Equations (4a) and (4b) were solved numerically by calculation left parts of these equations within some intervals of wavespeeds $c \in [c_1, c_2]$ and frequencies $\omega \in [2\pi f_1, 2\pi f_2]$ for given plate thickness 2*h* and wavespeeds of longitudinal $c_L$ and shear $c_T$ waves, which depend on the material properties. The calculation results establish only one zero-order anti-symmetric Lamb wave can be excited at the accepted frequencies. It was confirmed by the finite element simulation of the waves propagation using FE model described in the next section of this article.

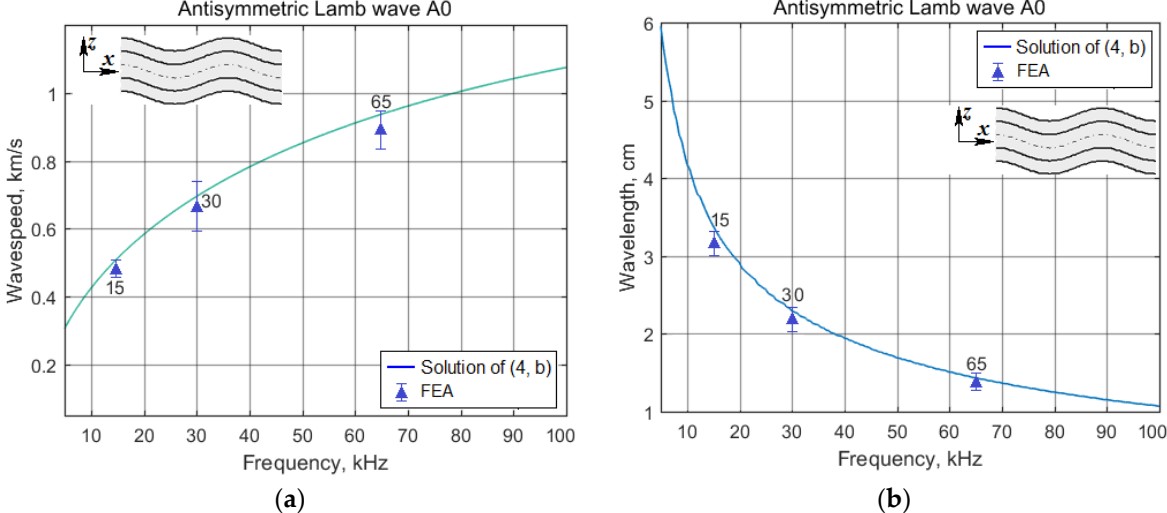

**Figure 5.** The dispersion curves for the wavespeed (**a**) and wavelength (**b**) of the Lamb wave A0 that can be excited in the studied panel at the frequencies 10–100 kHz together with the values of the wavespeeds and wavelenghts, which are computed by the postprocessing of FE simulation.

In order to analyze the frequency spectrum of horizontally polarized SH waves in the studied structure we used the frequency equation in the dimensionless form [1]

$$\Omega^2 = n^2 + \xi^2,\tag{6}$$

where the dimensionless frequency $\Omega$ and the dimensionless wavenumber $\xi$ are defined as

$$\Omega = \frac{2h\omega}{\pi c_T};\tag{7a}$$

$$\xi = \frac{2kh}{\pi} \ .\tag{7b}$$

Equation (6) yields an infinite number of continuous curves, called branches, each corresponding to an integer value of *n*. A branch displays the relationship between the dimensionless frequency $\Omega$ and the dimensionless wavenumber $\xi$ for a particular mode of propagation. Our calculations revealed only one symmetric zero-order mode SS0 of the horizontally polarized SH waves that can be excited in the panel at the accepted conditions. As one can see in Figure 6, the wavespeed (and wavelength, too) for this wave does not depend on the frequency, but because the solution of Equation (6) is highly dependent on the mechanical properties of the material where the wave propagates, we present in Figure 6 three wavespeeds, which correspond to the upper, middle, and lower values of the confidence interval for the Young's modulus of the panel's material. This plot also contains confidence intervals for the wavespeed, calculated from the FE simulation results for the three cases of modeled excitation frequencies.

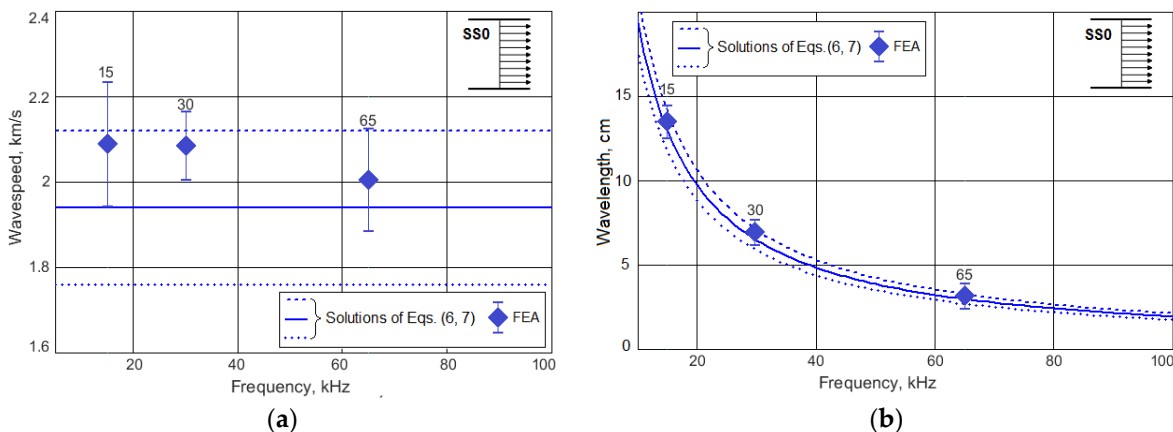

**Figure 6.** The dispersion curves for the wavespeed (**a**) and wavelength (**b**) of the horizontally polarized SS0 wave that can be excited in the studied panel at the frequencies 10–100 kHz together with the results of FE simulation.

The contemplation of Figures 5 and 6 suggests that A0 Lamb waves can be used for the detection of enough large imperfections in the tested structures in the whole excitation frequency range, whereas the horizontally polarized SS0 waves can be useful only at the higher frequencies of the considered frequency range. The features of the wave directivity at their excitation by the angle beam wedge actuator were studied in detail at the finite element simulation of the wave generation and propagation.

*3.2. Finite Element Analysis of Acoustic Wave Propagation in the Studied Quasi-Isotropic Panel*

The FE model, which was constructed in the Comsol Mutiphysics soft tool, copied the experimentally investigated situation presented in the Figure 3. All properties of the modeled parts were matched to the experimentally measured ones and identified as presented in Section 2 of this article. The geometry of the modeled system is presented in Figure 7.

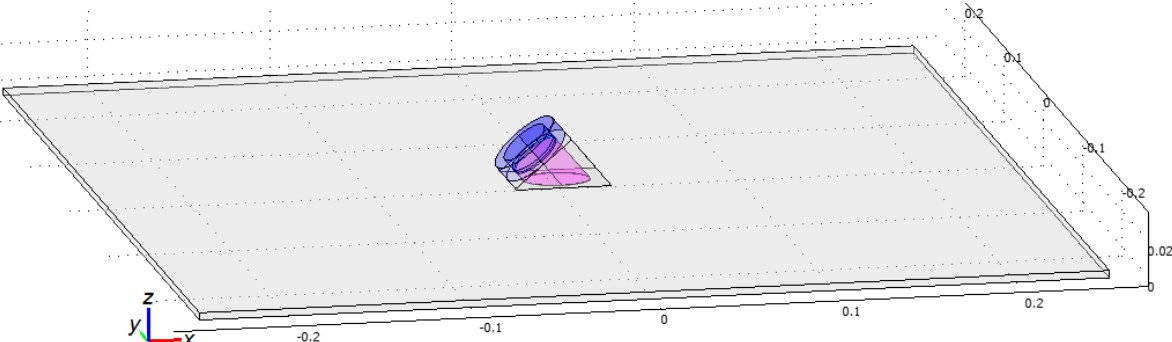

**Figure 7.** Geometry of the finite element model for the studied system.

The great advantage of the developed model is the ability to analyze phenomena that was not possible during experimental research. The wave generation and propagation process was considered by the transient analysis. The piezoelectric actuator was driven by the tone burst voltage signal described in Section 2.3. The simulated driven potential was always stopped before time instant when the wave reaches the edge of the plate. This eliminated the occurrence of reflection and distortion of the recorded sensor signals. The postprocessing abilities allowed to monitor the out-of-plane and in-plane displacements of the points on the panel surface independently.

The wavespeed along the radial lines was estimated by the time of flight—time lag between two synchronized signals of sensors, which were positioned at the distances 10, 15, or 17.5 cm from the center of the elliptic actuator's footprint. The choice of the optimal distance depended on the frequency

of excitation. Numerical processing of the files containing a time history of the process was carried out using the developed program module in MATLAB environment. To calculate the wavespeed, 4–5 time lags of waves in the signal were used, which made it possible to obtain the confidence intervals like those presented in Figures 5 and 6. The careful observation during some periods of oscillations of displacements in the slices, which are normal to the panel surface and oriented along the radial lines confirmed two types, lengths and speeds of propagating waves: Lamb A0 and horizontally polarized SS0. The discrepancies between the values of wavespeeds obtained on the base of FE simulation and those calculated from Equations (4–7), which use the experimentally studied plate's material properties, are enough small to confirm the validity of the presented results.

### 3.3. Analysis of Directivity for $A_0$ Lamb and SS0 Horizontally Polarized Waves Excited by the Angle Beam Wedge Actuator

In order to determine the waves directivity, each waves type was analyzed along the radial lines oriented at the angles 0°–180° during time intervals depicted in Figure 4. The angular step for the analyzed orientations was accepted to be 10 degrees. The virtual sensor's signal was allocated on the time interval, then its modulus was integrated to obtain its averaged value. Such averaging was necessary due to some distortions of waveform during its propagation. Most reliable results were obtained for the sine excitation signal with a smooth increase and stabilization of the amplitude, which was taken from the sensor located at a distance of 15 cm from the geometrical center of the panel. The normalized values of averaged out-of-plane (for A0 waves) and in-plane (for SS0 waves) displacements, which were calculated for 15, 30, and 65 kHz that correspond to the first three natural vibration modes of actuator are presented in Figure 8. For the Lamb wave A0 mode at the frequency 30 kHz the calculated directivity diagram is combined with the confidence intervals for the wave intensity, which were obtained after the numerical processing of experimental data.

The particular shapes of the waves intensity angular distributions can be explained by the forms of the actuator's vibration modes. The very large computational complexity of the task did not allow us to reveal the distribution of contact stress on the actuator footprint in the same way as was done in [31]. These stresses sharply increase near the border of the footprint and change around its perimeter. Reliable identification of these stresses requires an extremely fine finite element meshing, which could increase the task's degrees of freedom (DoF) above 1.5 million that corresponds to the very big operating memory. Therefore, the amplitude contact displacements at the used frequencies excitation were analyzed. The radial distributions of the out-of-plane and in-plane displacements amplitude under the footprint of actuator are presented in Figure 9, which demonstrates that angular distribution of the generated waves intensity (see Figure 8) follows the shape of the amplitudes of normal and radial displacements within the elliptic contact area. Plate excitation at the different eigenfrequencies of the actuator leads to the significant modification of the waves' directivity pattern. The well-defined directivity of the generated waves and their length of 1–2 cm make it possible to use them for identification of relatively large imperfections in the structure of the material, such as delamination, local change of the mechanical properties, inclusions, and another defects [6,7,9,10,17].

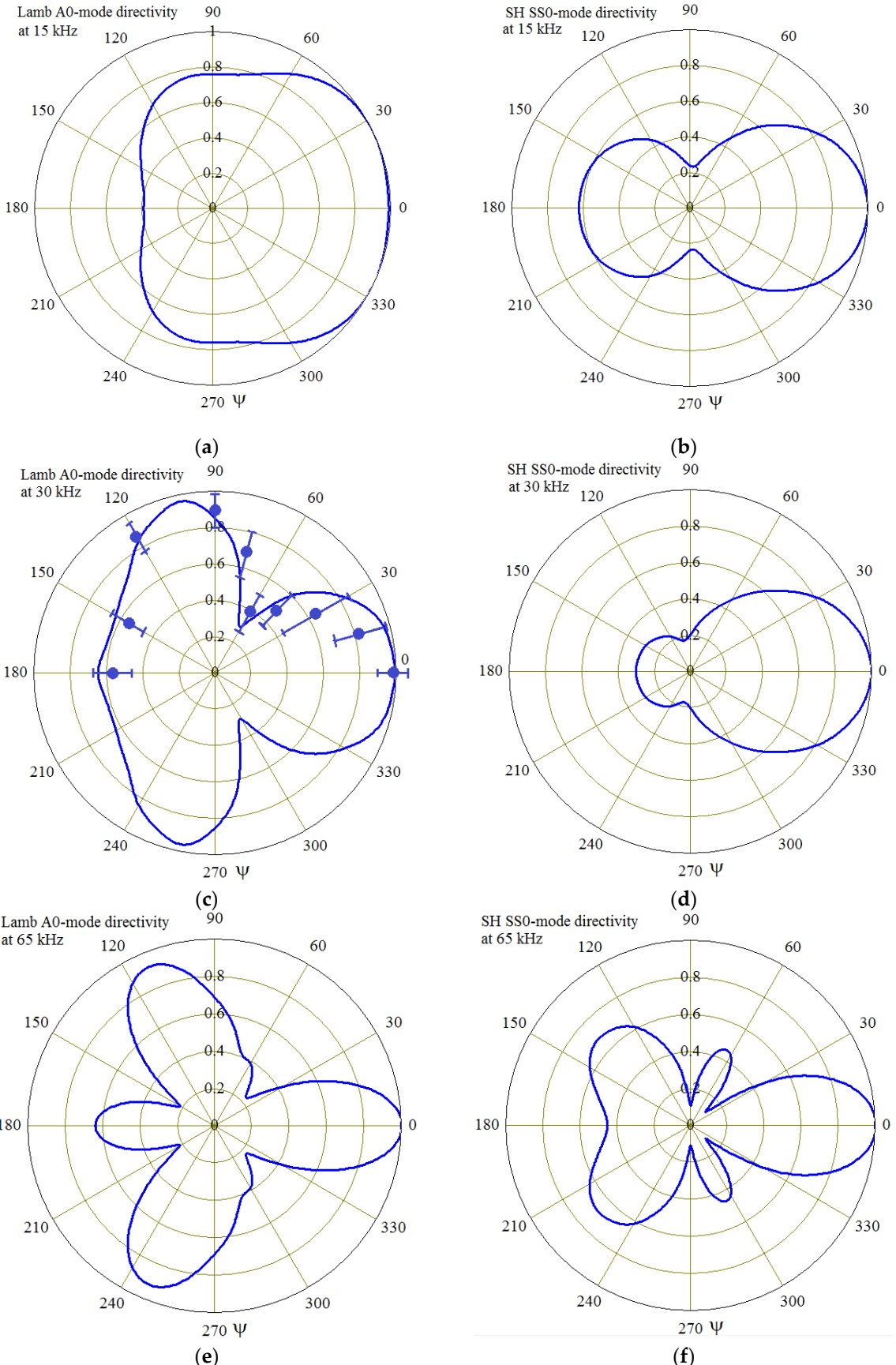

**Figure 8.** The normalized directivity diagrams for the Lamb A0 waves (**a**,**c**,**e**) and for the horizontally polarized SS0 waves (**b**,**d**,**f**) at the modeled eigenfrequencies of actuator: 15 (**a**,**b**), 30 (**c**,**d**) and 65 kHz (**e**,**f**).

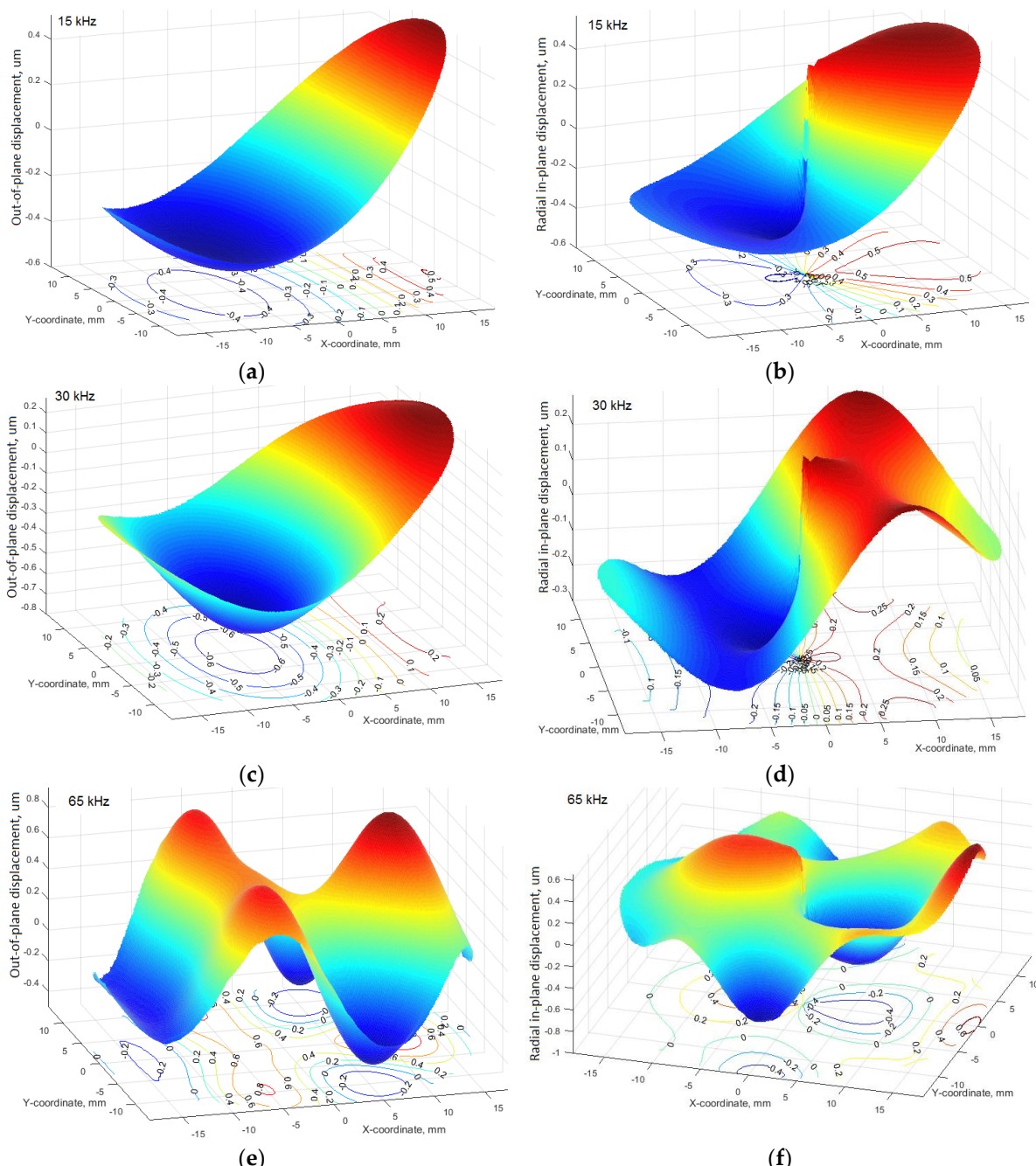

**Figure 9.** The radial distributions of the out-of-plane (**a**,**c**,**e**) and in-plane (**b**,**d**,**f**) amplitude displacements under footprint of angle beam wedge actuator exciting the studied plastic panel at the excitation frequencies 15 (**a**,**b**), 30 (**c**,**d**), and 65 kHz (**e**,**f**).

## 4. Conclusions

In order to improve the abilities of acoustic-based SHM and NDE techniques for the thin-walled structures made of plastic materials and polymeric composites, in particular the waves directivity, their propagation in a plastic panel excited by the angle beam wedge actuator were studied both experimentally and numerically. Our study was motivated by the impossibility to control the direction of the emitted waves at the use of well-known circular or annular piezoelectric transducers.

The dispersive analysis of the waves that can be excited in the studied plastic panel, was fulfilled using preliminary measured material properties. The found values allowed to find two kinds of the generated acoustic waves (anti-symmetric Lamb waves A0 and horizontally polarized SH waves SS0)

which both are zero-order modes. The presence of two types of waves generated by the investigated actuator in the low-frequency range, and their characteristics, are confirmed by the comparison of dispersion curves, finite element calculations, and experiments, which were performed under the studied plastic panel. The upper limit of the frequency range for our study was chosen due to the material damping that causes the very intensive frequency dependent attenuation of propagated waves. The chosen frequency range was the result of a compromise between the desire to obtain high defects resolution by generating short waves, their adjustable directivity and maximum propagation length.

The finite element model of the studied system includes the actuator's sub-model (see Figure 7), whose properties corresponded with great accuracy to the properties of a real actuator used in experiments. In order to identify the actuator's structure (dimensions of the parts, elastic, damping, and electro-mechanical properties) it was studied experimentally by using the measurement of its electric capacitance and frequency responses of the consumed electric current and the oscillation amplitude on the surface of elliptic contact. These results were used at the development of the finite element model of the transducer.

Our experiments with the real excited panel and numerical simulation of its finite element prototype convincingly demonstrated the satisfactory directivity of the actuated waves at their excitation on the frequencies that corresponded to the natural modes of the actuator oscillation. The considered results suggest the possibility to efficiently use the angle beam wedge transducers in the SHM of thin-walled structures made of highly damped materials. Their efficiency can be provided due to the ability of beamsteering of excitation waves that is unavailable to the circular, annular, and other types of omnidirectional transducers [12,13,22,32]. However, the efficient use of the proposed method for other analyzed materials seems possible only after conducting a study similar to that presented in this article. It should be noted that the reported technique, which was applied to the simple enough quasi-isotropic plastic, relates to the initial stage of a project aiming to develop controlled acoustic monitoring of high loaded structures made of materials with orthotropic anisotropy, in particular, glass- and carbon-reinforced plastics.

**Author Contributions:** Conceptualization, E.K. and S.S.; methodology, S.S.; experimental investigation, V.C. and E.R.; software development, M.S.; FE models development and validation, S.S.; results numerical processing, V.C. and M.S.; writing—original draft preparation, review and editing, S.S.; supervision and project administration, E.K.; funding acquisition, S.S. and E.K.

**Funding:** This research was funded by the German Federal Ministry of Education and Research (BMBF), grant 13FH009IX5, and by the Federal Research Center "The Southern Center of Russian Academy of Science", project A16-116012610052-3.

**Acknowledgments:** The authors wish to acknowledge the valuable technical support from the Institute of Mechanics and Applied Mathematics of the Southern Federal University directed by M. Karyakin who provided the Acoustic Research Lab for the experimental research.

**Conflicts of Interest:** The authors declare no conflict of interest.

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
