# Peer review of "On the Directivity of Acoustic Waves Generated by the Angle Beam Wedge Actuator in Thin-Walled Structures"

_actuators, doi:10.3390/act8030064_

Round 1

Reviewer 1 Report

Please provide more information about the thin-walled plates under test. What was the ratio of wall thickness to overall plate thickness? What materials comprised the structure of the panel?

The purpose of Chapter 4 is not quite clear. Results of the work are given in preceding chapters, and the discussion is very scarce. I would recommend to exclude this chapter, moving its contents in part to the chapters presenting results and in part to the Conclusion chapter.

Please check English and get rid of numerous grammatical errors in the text.

Line 229. It should be Figs 4,a and 4,b, not 3,a,b. 

Author Response

Response to Reviewers

The authors would like to thank the reviewers for their constructive and helpful comments, which have improved the quality of the manuscript. The changes have been highlighted in the text by yellow color.

Reviewer 1

Please provide more information about the thin-walled plates under test. What was the ratio of wall thickness to overall plate thickness? What materials comprised the structure of the panel?

The studied plastic plate has the a constant thickness of 4 mm over its entire area. Information of the plate material is included in the paper text.

The purpose of Chapter 4 is not quite clear. Results of the work are given in preceding chapters, and the discussion is very scarce. I would recommend to exclude this chapter, moving its contents in part to the chapters presenting results and in part to the Conclusion chapter.

The reviewer is absolutely right. The paper structure is reorganized according to his very useful advice. Chapter 4 is excluded; its content is moved to sections 3 and Conclusions, and text is substantially corrected.

Please check English and get rid of numerous grammatical errors in the text.

Many grammatical errors in the text are corrected. They are highlighted in the paper text.

Line 229. It should be Figs 4,a and 4,b, not 3,a,b. 

This error, which occurs twice in the text, is fixed.

Reviewer 2 Report

The paper is devoted to complex investigation of the characteristics of acoustic waves generated by standard angle-beam wedge actuator in a thin walled plastic panel. Previous works of other authors cited in paper were devoted to analogous investigations but in thick metal part. This paper is important from practical point of view and will be useful for researcher who involved in NDE of layered composites and plastic materials. Paper is very detailed and very well organized.

Some remarks

In Abstract it is necessary use full term. It is necessary to decript SHM in Abstract. Line 126 Instead S0 should be SS0.

Author Response

Response to Reviewers

The authors would like to thank the reviewers for their constructive and helpful comments, which have improved the quality of the manuscript. The changes have been highlighted in the text by yellow color.

Reviewer 2

1.The paper is devoted to complex investigation of the characteristics of acoustic waves generated by standard angle-beam wedge actuator in a thin walled plastic panel. Previous works of other authors cited in paper were devoted to analogous investigations but in thick metal part. This paper is important from practical point of view and will be useful for researcher who involved in NDE of layered composites and plastic materials. Paper is very detailed and very well organized.

The authors are pleased for the estimation and grateful for the careful analysis of the manuscript.

In Abstract it is necessary use full term. It is necessary to decrypt SHM in Abstract.

The explanation of abbreviations is included in Abstract and highlighted.

Line 126 Instead S0 should be SS0. 

Thank you very much for this remark.
